# Interstitials in f.c.c. High Entropy Alloys

**Ian Baker** 

Thayer School of Engineering, Dartmouth College, Hanover, NH 03755, USA; Ian.Baker@Dartmouth.edu

**Abstract:** The effects of interstitials on the mechanical properties of single-phase f.c.c. high entropy alloys (HEAs) have been assessed based on a review of the literature. It is found that in nearly all studies, carbon increases the yield strength, in some cases by more than in traditional alloys. This suggests that carbon can be an excellent way to strengthen HEAs. This strength increase is related to the lattice expansion from the carbon. The effects on other mechanical behavior is mixed. Most studies show a slight reduction in ductility due to carbon, but a few show increases in ductility accompanying the yield strength increase. Similarly, some studies show little or modest increases in work-hardening rate (WHR) due to carbon, whereas a few show a substantial increase. These latter effects are due to changes in deformation mode. For both undoped and carbon doped CoCrFeMnNi, the room temperature ductility decreases slightly with decreasing grain size until ~2–5 μm, below which the ductility appears to decrease rapidly. The room temperature WHR also appears to decrease with decreasing grain size in both undoped and carbon-doped CoCrFeMnNi and in nitrogen-doped medium entropy alloy NiCoCr, and, at least for the undoped HEA, shows a sharp decrease at grain sizes <2 μm. Interestingly, carbon has been shown to almost double the Hall–Petch strengthening in CoCrFeMnNi, suggesting the segregation of carbon to the grain boundaries. There have been few studies on the effects of other interstitials such as boron, nitrogen and hydrogen. It is clear that more research is needed on interstitials both to understand their effects on mechanical properties and to optimize their use.

**Keywords:** high entropy alloys; interstitials; lattice expansion; mechanical properties; CoCrFeMnNi; Hall–Petch slope

## 1. Introduction

The idea of multi-component or multi-element alloys with equal atomic proportions that could be single phase was separately introduced by Cantor et al. [1] and Yeh et al. [2]. Yeh et al. [2] coined the name High Entropy Alloys (HEAs) for such alloys based on the concept that a high configurational entropy, $\Delta S_{\text{conf}}$, could offset the enthalpy of intermetallic phase formation. Yeh et al. [2] defined HEAs as alloys containing at least five elements with concentrations from 5 to 35 atomic percent, although many subsequent works have referred to five component alloys with some components less than 5 at. % or even to four component alloys as HEAs even though they do not fit the original definition.

Yeh et al. [2] also predicted that while single phase at high temperature, $T$, where the entropy term, $T\Delta S_{\text{conf}}$, is more important, HEAs could precipitate second phases or undergo spinodal decomposition at low temperature, where the configurational entropy term is of less importance. Such behavior has been noted in some f.c.c. HEAs such as the so-called Cantor alloy, $Fe_{20}Cr_{20}Mn_{20}Ni_{20}Co_{20}$ [3] and $Fe_{40.4}Ni_{11.3}Mn_{34.8}Al_{7.5}Cr_6$ (in at. %) [4], that are single phase f.c.c. upon casting but in which other phases precipitate after annealing. Indeed, this may be the case in many HEAs given long-term anneals at low or intermediate temperatures.

Interestingly, even in these early studies the b.c.c. HEAs were shown to have a high yield strength, $\sigma_y$, [2] whereas the f.c.c. HEAs were relatively weak [1,2]. This feature has been borne out in many

later studies. One way to increase $\sigma_y$ for f.c.c. HEAs is by interstitial strengthening, an area in which there is growing activity.

The aim of this paper is to analyze the effect of interstitials on the mechanical properties of f.c.c. HEAs from data reported in the literature.

## 2. Methodology

We compare several parameters that describe the quasi-static mechanical properties, viz., either the yield stress, $\sigma_y$, or (when a range of grain sizes have been measured) the lattice resistance, $\sigma_0$; the elongation to failure, $\varepsilon_f$; the Hall–Petch slope, $k$; and the work-hardening rate, WHR.

There are several approaches that can be used to determine the effect of interstitials on the lattice resistance, $\sigma_0$, of a HEA. One way is to measure the critical resolved shear stress of HEA single-slip-oriented single crystals with and without the interstitial of interest, preferably with crystals of the same orientation. Thus far, there have been no studies published using this approach.

Second, one could use polycrystals, in which case $\sigma_y$ is given by

$$\sigma_y = \sigma_0 + kd^{-1/2}$$

where $d$ is the grain size. An interstitial can affect both $\sigma_0$ and $k$, but they are not always both increased by an interstitial addition. For instance, in the ordered f.c.c. compound $Ni_3Al$, boron increases $\sigma_0$ but reduces $k$ [5–7] while in the ordered b.c.c. compound FeAl, boron increases $k$ but appears to have no effect on $\sigma_0$ [8]—a similar phenomenon occurs for sulfur in nickel [9].

This gives us two ways to utilize polycrystalline specimens. First, one can measure $\sigma_y$ for different values of $d$ for the HEA both with and without the interstitial. This provides not only a value for $\sigma_0$, but also one for $k$. There have been no studies in which a HEA with and without an interstitial have been undertaken—although studies have been conducted on the medium entropy alloys NiCoCr with and without carbon [10] or nitrogen [11] and FeCoNiCr with and without nitrogen [12]. Separate groups have measured $\sigma_0$ and $k$ for a HEA with or without an interstitial and these data are compared here to obtain the interstitial strengthening effect. We should note that there are at least five effects that one needs to be aware of when making such a comparison.

(1) Even the undoped HEA may contain small amounts of interstitials introduced during processing or from the original elemental ingredients and these may have a significant effect, particularly on $k$—for instance, small amounts of sulfur (1–22 p.p.m.) introduced inadvertently into nickel castings [9] produced values of $k$ from 0.3 to 0.6 $MNm^{-1/2}$.

(2) Texture can affect $k$ [13].

(3) The value of $k$ affects the value of $\sigma_0$. Consider Figure 1 in which two materials (or the same material processed in different ways) show the same $\sigma_y$ for large-grained specimens (arrowed on the figure). Because they have different $k$ values, they have quite different values of $\sigma_0$. Thus, $\sigma_0$ is not the same as $\sigma_y$ for a large-grained specimen because $\sigma_0$ is obtained by extrapolation (the dotted lines on Figure 1).

(4) Different authors use different threshold misorientations to determine the grain size from electron backscatter diffraction data obtained in a scanning electron microscope, e.g., some use 5° misorientation while some use 15°, which can result in quite different grain sizes.

(5) Schneider et al. [14] recently showed that for the MEA NiCoCr that if twin boundaries are included in the Hall–Petch analysis that the $k$ values can be lower (~600 MPa·$\mu m^{-1/2}$ versus ~820 MPa·$\mu m^{-1/2}$ at 273 K), although the temperature dependence of $k$ is similar.

Second, one could compare $\sigma_y$ of an HEA with and without interstitial atoms for grain sizes that were sufficiently large so that the $kd^{-1/2}$ term is negligible. Grain sizes of $\geq 100$ $\mu m$ typically satisfy this criterion [13]. This approach has been performed by some researchers on HEAs as described below.

Microhardness measurements of a HEA with and without an interstitial will also provide a good measure of the lattice strength as long as the distance to the grain boundaries from the indentation is

2.5–3 times the indentation size (ISO 6507-1). $\sigma_y$ can be assessed for a perfectly elastic/perfectly plastic solid by dividing the hardness by three. However, since real alloys rarely conform to be a perfectly elastic/perfectly plastic solid, the hardness corresponds to the flow stress at about 8–10% strain [15].

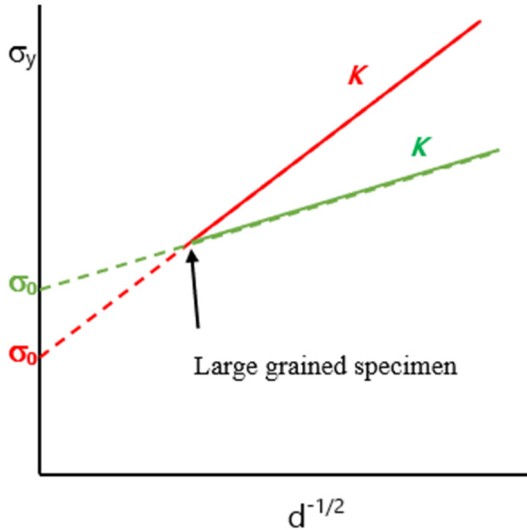

**Figure 1.** Schematic Hall–Petch plot for two materials that have the same yield strength, $\sigma_y$, for large-grained samples but different Hall–Petch slopes, *k*, leading to different values of the lattice resistance, $\sigma_0$.

In most of the papers reviewed here, tensile test data is presented only as engineering stress, $\sigma_e$, versus engineering strain, $\varepsilon_e$. Thus, we report $\sigma_y$ or $\sigma_0$ and $\varepsilon_f$; as engineering stresses and engineering strain, respectively, rather than as the true stress, $\sigma_T$, and true strain, $\varepsilon_T$. However, conventionally the WHR is defined as ($\delta\sigma_T/\delta\varepsilon_T$). Thus, to calculate WHR, we used ($\sigma_{UTS} - \sigma_y$)(1 + $\varepsilon_f$)/ln(1 + $\varepsilon_f$), where $\sigma_{UTS}$ is the ultimate tensile strength, and the term ln(1 + $\varepsilon_e$) is $\varepsilon_T$.

Finally, when analyzing data, it is important to note that in some papers the interstitials have been measured after casting the HEA, whereas in other cases the interstitial content (and that of the other elements) was assumed to be the same as that added to the melt. We specifically note the cases where the composition has been measured. The latter not only does not allow for loss or pick up of the interstitial during processing, but may not account for interstitials arising from the elemental ingredients of the HEA. Thus, the calculated strengthening effect for the latter materials clearly has some inherent errors.

Below, we first analyze the effects of carbon on the mechanical properties of different HEAs and then discuss works that have added nitrogen, boron or hydrogen to various HEAs.

## 3. Interstitial Strengthening

Most work on interstitials in f.c.c. HEAs has been on carbon doping and on equiatomic CoCrFeMnNi. In all HEA studies, the carbon was introduced by dissolving in the melt, sometimes as a carbide, rather than by carburization. The interstitials are assumed to be completely in solution unless otherwise noted. Compositions are given in at. % unless otherwise noted.

### 3.1. Carbon Doping—CoCrFeMnNi

Two studies have determined the critical resolved shear stress (CRSS) of single crystals of equiatomic, single-phase f.c.c. CoCrFeMnNi. Patriarca et al. [16] determined the CRSS at 77 and 273 K to be 175 MPa and 70 MPa, respectively, from compression tests on (591)-oriented single crystals. Later, Abuzaid and Sehitoglu [17] determined the CRSS for several crystal orientations under tension at 77

and 293 K to be 145–172 MPa and 53–60 MPa, respectively. Thus far, there have been no studies on similar HEAs containing interstitials.

There have been a number of studies looking at grain-size strengthening in cast, cold-rolled, and recrystallized polycrystalline equiatomic CoCrFeMnNi. Otto et al. [18] determined $\sigma_y$ as a function of grain size for equiatomic single-phase f.c.c. CoCrFeMnNi at temperatures from 77 to 1073 K, see Tables 1 and 2. Note that only three grain sizes were tested (4.4, 50, 155 μm) at each temperature with the reported results being the average of two tensile tests performed at each condition. $k$ was found to decrease gradually from 538 MPa·μm$^{-1/2}$ at 77 K to 425 MPa·μm$^{-1/2}$ at 473 K, was independent of temperature from 473 K to 873 K, and then decreased rapidly to 127 MPa·μm$^{-1/2}$ at 1073 K, see Figure 2. In contrast, $\sigma_0$ decreased rapidly from 310 MPa at 77 K to 125 MPa at 293 K, but then decreased more slowly to 43 MPa at 873 K, see Figure 3.

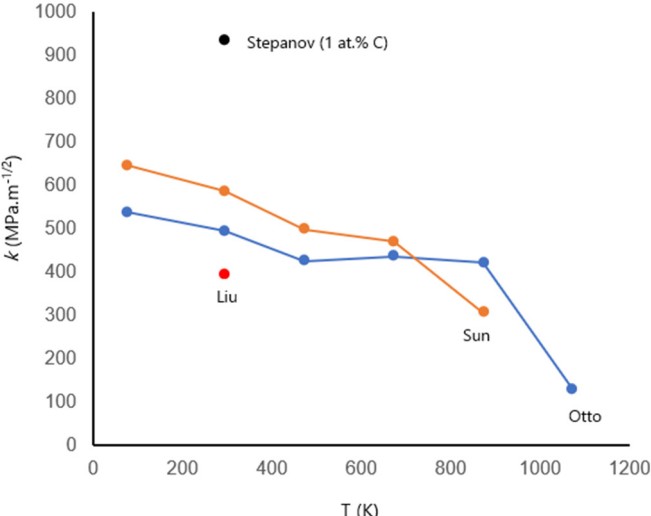

**Figure 2.** Graph of Hall–Petch parameter, *k*, versus temperature for undoped equiatomic CoCrFeMnNi, data from [18–20]; and for 1 at. % C-doped CoCrFeMnNi from Stepanov et al. [21].

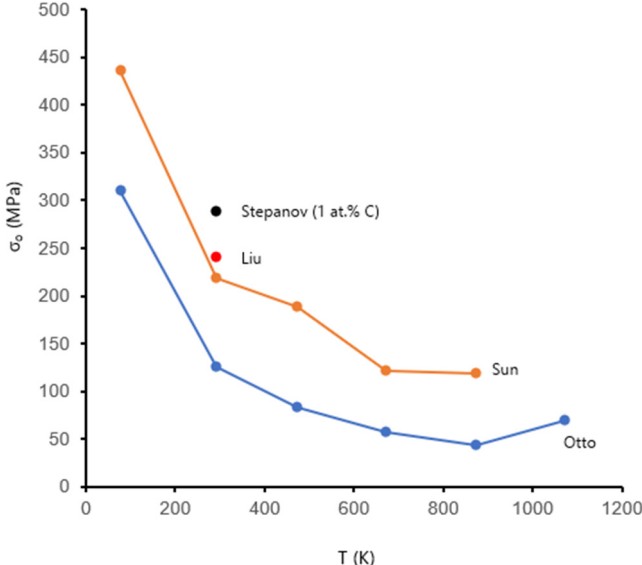

**Figure 3.** Graph of lattice resistance, $\sigma_0$, versus temperature for undoped equiatomic CoCrFeMnNi, data from [18–20]; and for 1 at. % C-doped CoCrFeMnNi from Stepanov et al. [21].

Interestingly, Otto et al. [18] reported a higher $\sigma_0$ value of 69 MPa at 1073 K than at 673 K or 873 K. This does not indicate an error in measurements; $\sigma_y$ for the largest grain-sized material always

decreased with increasing temperature, and was 93 MPa, 79 MPa and 74 MPa at 673, 873 and 1073 K, respectively. In fact, the higher value of $\sigma_0$ at 1073 K compared to 673 K and 873 K simply reflects the lower value of *k* at that temperature.

WHR showed a similar dependence on temperature. At 77 K, the WHR was independent of grain size at ~1725 MPa. The WHR decreased continuously with increasing temperature. At room temperature the two largest grain-sized (50 μm, 155 μm) HEA showed a slightly higher WHR than the finest grain-sized (4.4 μm) material, i.e., ~1260 MPa versus ~1100 MPa. The finest grained HEA showed no work-hardening at 1073 K.

$\varepsilon_f$ decreased continuously with increasing temperature for large-grained (155 μm) CoCrFeMnNi from 84% at 77 K to 20% at 1073 K, whereas the finest-grained (4.4 μm) material showed a decrease from 72% at 77 K to 32% at 673 K after which it increased up to 51% at 1073 K. $\varepsilon_f$ appeared to be lowest for the finest grain size from 77 to 873 K, whereas fine-grained material showed double the $\varepsilon_f$ (~50%) at 1073 K compared to the two coarser-grained materials. Room temperature specimens showed average $\varepsilon_f$ values of 59% for the largest grain sizes (50 μm, 155 μm), and 51% for the finest-grained (4.4 μm) material. Deformation occurred by planar dislocation glide at low strains, but that (nano)twinning occurred at higher strains, leading to the high WHR and large observed $\varepsilon_f$ values [18,22]. Deformation nanotwinning occurs at lower strains as the temperature is decreased [22] (~7.4% at 77 K and ~25% at 293 K) since the critical stress for nanotwinning is reached at a lower strain [23].

Sun et al. [19] also determined the Hall–Petch behavior of equiatomic single-phase f.c.c. CoCrFeMnNi from 77 to 873 K. The cast material was hot-forged and either annealed (to produce coarse-grained material) or cold-rolled and recrystallized (to produce fine-grained material). As with Otto et al. [18], only three grain sizes were tested (0.65, 2.1 and 105 μm) at each temperature, but with only one test appearing to have been performed for each grain size. They found that *k* decreased roughly linearly from 645 MPa·μm$^{-1/2}$ at 77 K to 306 MPa·μm$^{-1/2}$ at 873 K, see Tables 1 and 2, and Figure 2. It is worth noting that their *k* values were greater than those determined by Otto et al. [18] at all temperatures up to 673 K, but less at 873 K—they did not observe the plateau in *k* values from 473 to 873 K reported by Otto et al. [18]. One should be somewhat wary of ascribing too much significance to these differences in *k* between the two groups since, as noted earlier, changes in texture due to differences in processing conditions and small changes in interstitial content (even in these undoped HEA) can each have large effects on its value.

Similar to the results of Otto et al. [18], Sun et al. [19] found that $\sigma_0$ decreased rapidly from 436 MPa at 77 K to 188 MPa at 293 K, but then decreased more slowly to 119 MPa at 873 K. Their $\sigma_0$ values were always significantly greater than those determined by Otto et al. [18] at all temperatures, see Tables 1 and 2, and Figure 3.

At 77 K, the WHR was significantly lower for the finest grain-sized (0.65 μm) HEA at 854 MPa compared to the two larger grain-sized (2.1 μm, 105 μm) HEAs at 1644 MPa and 1465 MPa, respectively. This grain size dependence of the WHR is even greater at 293 K, i.e., 287, 1115 and 1168 MPa. The value for the largest grain size at 293 K is similar to that calculated from Otto et al. [18]. Interestingly, the two finest grain sized (0.65 μm, 2.1 μm) materials showed no work-hardening at 873 K.

Sun et al. [19] found that $\varepsilon_f$ decreased with increasing temperature for the two finer grain sizes e.g., from 41% at 77 K to 2% at 873 K for the 0.65 μm material, but for the coarsest-grained (105 μm) material $\varepsilon_f$ decreased from 82% at 77 K to ~60% at 203 K after which it was constant up to 293–873 K. The coarser the grain size the greater the $\varepsilon_f$ value at all temperatures, e.g., at room temperature $\varepsilon_f$ was 27%, 33% and 60% for grain sizes of 0.65 μm, 2.1 μm and 105 μm, respectively.

Sun et al. [19] confirmed the earlier observations [18,22] that deformation occurs by a combination of dislocation motion and twinning at 77 K and 293 K. Twinning did not occur at higher temperatures, but dynamic recrystallization was observed at 873 K.

Recently, Liu et al. [20] determined the Hall–Petch behavior of equiatomic single-phase f.c.c. CoCrFeMnNi at room temperature only. The cast material was subsequently cold-rolled and recrystallized. They obtained values for $\sigma_0$ and *k* of 214 MPa and 394 MPa·μm$^{-1/2}$, respectively,

see Table 1 and Figures 2 and 3. We should note that in this study four different grain sizes were tested but they spanned less than an order of magnitude (3.9 μm, 10.8 μm, 20.5 mm and 30.1 μm) and only one test appears to have been performed for each grain size. Their room temperature value of $\sigma_0$ is almost identical to that obtained by Sun et al. [19], but their *k* value is less than that of both Sun et al. [19] and Otto et al. [18].

**Table 1.** Values of the Hall–Petch parameters at different temperatures for CoCrFeMnNi. From Otto et al. [18], Sun et al. [19], and Liu et al. [20].

| *T* (K) | $\sigma_0$ (MPa) Otto et al. [18] | $\sigma_0$ (MPa) Sun et al. [19] | $\sigma_0$ (MPa) Liu et al. [20] | *k* (MPa·μm$^{-1/2}$) Otto et al. [18] | *k* (MPa·μm$^{-1/2}$) Sun et al. [19] | *k* (MPa·μm$^{-1/2}$) Liu et al. [20] |
|---|---|---|---|---|---|---|
| 77 | 310 | 436 | - | 538 | 645 | - |
| 293 | 125 | 218 | 214 | 494 | 586 | 394 |
| 473 | 83 | 188 | - | 425 | 497 | - |
| 673 | 57 | 121 | - | 436 | 469 | - |
| 873 | 43 | 119 | - | 421 | 306 | - |
| 1073 | 69 | - | - | 127 | - | - |

The WHR calculated from data in Figure 6 in Liu et al. [20] showed that the room temperature WHR was independent of grain size at 1100–1166 MPa.

$\varepsilon_f$ also showed little grain size dependence but the finest grain size (3.9 μm) showed the lowest $\varepsilon_f$ of 51% while the largest grain size (30.1 μm) showed the largest $\varepsilon_f$ of 55%, with the intermediate grain-sized specimens (10.8 mm, 20.5 μm) showing an intermediate $\varepsilon_f$ of 53%. The tensile specimens were produced from cold-rolled and annealed, recrystallized material. They showed that the HEA had considerable segregation both before and after the recrystallization anneals. Since the different grain sizes were produced by anneals at different temperatures (1073–1273 K), presumably the segregation varied for different grain sizes.

Earlier, Liu et al. [24] demonstrated a Hall–Petch relationship between the Vickers hardness and the grain size for equiatomic CoCrFeMnNi at 293 K and obtained a *k* value of 677 MPa·μm$^{-1/2}$, which is higher than the values obtained by Liu et al. [20] of 394 MPa·μm$^{-1/2}$, Otto et al. [18] of 494 MPa·μm$^{-1/2}$ and Sun et al. [19] of 586 MPa·μm$^{-1/2}$. The higher value may reflect that the Vickers hardness represents the flow stress at 8–10% strain rather than $\sigma_y$.

Stepanov et al. [21] determined $\sigma_y$ as a function of grain size at 293 K for as-cast equiatomic CoCrFeMnNi containing (measured) 0.97 at. % C that was given a homogenization anneal, 80% cold-rolled and annealed at temperatures from 873 to 1373 K. Specimens annealed at temperatures ≥973 K were fully recrystallized and occasionally contained a very small volume fraction (≤0.2%) of $M_{23}C_6$ particles within grains that would likely have had a negligible effect on the mechanical properties. They compared their Hall–Petch plot to that of Otto et al. [18] at the same temperature for C-free CoCrFeMnNi. They performed three tensile tests on each of four grain sizes, i.e., 1.4, 4.9, 12 and 69.7 μm. We should note that the finest-grained material (1.4 μm), which was annealed at 700 °C, had a very inhomogeneous grain structure consisting of regions of very fine grains and regions of much coarser grains (Figure 5b in their paper). This resulted in the $\sigma_y$ values for this material showing a very large scatter, i.e., the average $\sigma_y$ was 1070 MPa, but the individual values were 1010, 1050 and 1150 MPa [25].

The carbon addition increased the room temperature value of *k*, from 394 to 586 MPa·μm$^{-1/2}$ (measured by [18–20]) to 935 MPa·μm$^{-1/2}$. Thus, carbon has a strong effect on the Hall–Petch slope, see Figure 2. The carbon addition more than doubled the room temperature $\sigma_0$ from the 125 MPa determined by Otto et al. [18] to 288 MPa, a strengthening effect due to carbon of 168 MPa/at. % C. If, however, we compare the $\sigma_0$ value from Stepanov et al. [21] with the lattice friction value from

Sun et al. [19] of 218 MPa, the strengthening effect due to carbon is only 72 MPa/at. % C, see Figure 3. It is worth noting that although there were a few particles present in the carbon-doped CoCrFeMnNi, Stepanov et al. [21] noted that their wide separation meant that they contributed little to the strength.

The WHR calculated from Table 1 in Stepanov et al. [21] suggests a grain size dependence, i.e., 1131, 1170 and 1408 MPa for 4.9, 12 and 69.7 μm grain-sized HEAs. The large scatter of $\sigma_y$ values for the fine-grained (1.4 μm) material makes calculating the WHR problematic for this grain size, see above.

Stepanov et al. [21] found a very large grain-size dependence for $\varepsilon_f$ with it increasing from 14% for 1.4 μm grains to 37% for 4.9 μm grains, to 48% for 12 μm grains, to 66% for 69.7 μm grains, a trend similar to that noted by Sun et al. [19]. The lowest $\varepsilon_f$ value was less than any value recorded by Sun et al. [19], Otto et al. [18] or Liu et al. [20]—a feature that may be related to the inhomogeneous microstructure of the finest grained material noted above—but the largest $\varepsilon_f$ had been observed by some of these other workers. The carbon appears to result in a greater dislocation density and fewer twins than in the undoped alloys, a feature that the authors ascribed to an increase in the stacking fault energy (SFE): ab initio quantum-mechanical calculations indeed indicate that carbon increases the SFE of CoCrFeMnNi [26].

**Table 2.** Lattice resistance, $\sigma_0$, from yield strength-grain size plots, and yield strength, $\sigma_y$, (both in MPa) for large grained equiatomic CoCrFeMnNi with and without carbon (at. % C indicated in parentheses) from the papers indicated.

| T(K)/Reference | $\sigma_0$ Otto et al. [18] | $\sigma_0$ Sun et al. [19] | $\sigma_0$ Liu et al. [20] | $\sigma_0$ Stepanov et al. [21] | $\sigma_y$ Wu et al. [27] | $\sigma_y$ Wu et al. [27] | $\sigma_y$ Chen et al. [28] | $\sigma_y$ Chen et al. [28] | $\sigma_y$ Li [29] |
|---|---|---|---|---|---|---|---|---|---|
| 77 | 310 | 436 | - | - | 350 | 510 (0.5C) | - | - | - |
| 293 | 125 | 218 | 214 | 288 (1C) | 165 | 225 (0.5C) | 250 | 310 (1.1C) | 200 (0.00, 0.25, 0.53, 0.9) |

There have been a number of studies where carbon has been added to large-grained equiatomic CoCrFeMnNi to assess its strengthening effect. Wu et al. [27] examined the effect of adding 0.5 at. % C on the strength at both 77 K and 293 K of large-grained (115 μm) CoCrFeMnNi that was given a homogenization anneal, cold-rolled and recrystallized and compared their results to those of Otto et al. [18]. The 0.5 at. % C increased $\sigma_y$ from 165 to 225 MPa at 293 K and from 350 to 510 MPa at 77 K, that is a strengthening effect due to C of 120 MPa/at. % C at 293 K and a much larger value at 320 MPa/at. % C at 77 K, see Table 2. The carbon addition decreased the $\varepsilon_f$ at both temperatures from ~85% to 69 % at 77 K, and from ~60 % to 38 % at 293 K.

The WHR, calculated from data in Table 1 in Wu et al. [27], showed that carbon increased the WHR substantially (compared to data in Otto et al. [18]) at both 77 K and 293 K to 1868 MPa (33% increase) and 1842 MPa (57% increase), respectively. The authors suggested that the increased WHR was due to the earlier onset of deformation twinning than in the undoped HEA, as suggested by electron backscattered diffraction images. This is in sharp contrast to the transmission electron microscope observations of Stepanov et al., who reported the opposite effect of carbon [21].

Chen et al. [28] found the (measured) 1.1 at. % C that they added to as-cast large-grained (110 μm—the authors did not state the grain size, thus, this value was obtained using the linear intercept method to estimate the grain size from Figure 10a of their paper) CoCrFeMnNi increased $\sigma_y$ from 250 to 310 MPa, while also slightly increasing $\varepsilon_f$ from 52% to 60%, see Table 2. Interestingly, the WHR did not show any effect of carbon with a value of 908–919 MPa, which is slightly lower than the values reported for large-grained undoped CoCrFeMnNi reported above. Note that the strength of their undoped CoCrFeMnNi was comparable to those reported by Otto et al. [18], Sun et al. [19] and Liu et al. [20], for coarser-grained materials. Chen et al.'s [28] carbon-doped CoCrFeMnNi had significant segregation of the carbon to interdendritic regions in the columnar grain structure—note that most other researchers gave cast material a homogenization anneal, and sometimes cold-rolling followed by a recrystallization anneal. The observed strengthening due to carbon was only 55 MPa/at.

% C. This lower value than those observed by Stepanov et al. [21] and Wu et al. [27] may be because of differences in grain size between Chen et al.'s [28] undoped and C-doped material. Similar to Wu et al. [27], Chen et al. [28] found that the carbon addition led to a greater extent of deformation twinning.

Li [29] looked at the effects of different carbon contents (0.25, 0.53, 0.9 at. %, measured by wet chemical analysis after casting) on the room-temperature mechanical properties of equiatomic CoCrFeMnNi at 293 K, which were found to be large elongated 200 μm-wide single-phase f.c.c. grains upon casting. After hot-rolling and a homogenization anneal at 1473 K, 200 μm equi-axed grains were also present. The elemental distribution after the latter treatment was homogeneous whereas it was inhomogeneous upon casting. They tested three specimens for each condition. Surprisingly, $\sigma_y$ of both the as-cast and thermo-mechanically-treated CoCrFeMnNi was ~200 MPa irrespective of carbon content, i.e., there appeared to be no solid solution strengthening from the carbon, or heat treatment, see Table 2.

In contrast, increasing the carbon content appeared to slightly increase the WHR for both the as-cast and thermo-mechanically-treated CoCrFeMnNi (calculated from Figure 9 [29]) from 829 MPa for 0.25 at. % C to 878 MPa for 0.53 at. % C to 1022 MPa for 0.9 at. % C for the as-cast HEA, and from 1108 MPa for 0.25 at. % C to 1103 MPa for 0.53 at. % C to 1284 MPa for 0.9 at. % C for the thermo-mechanically treated HEA. Note that the thermo-mechanically-treated CoCrFeMnNi had a higher WHR than the as-cast HEA for the same carbon content. However, it is worth noting that these values are all low compared to those reported by other researchers.

The carbon content and heat treatment also had little effect on $\varepsilon_f$ (44–51%); as noted by others, deformation at low strains was due to dislocation slip whereas twinning occurred at higher strains; the twinning density decreased as the carbon content increased, an effect ascribed to an increase in the SFE. These HEAs were also subjected to cold-rolling and annealing after which they had higher $\sigma_y$ values with reduced $\varepsilon_f$ for the larger carbon contents. It is not possible to assess the effects of interstitial carbon in the latter specimens since nanocarbides were present, and either the grain size was substantially reduced (4.5 μm) for the 0.2 at. % HEA, or they were not fully recrystallized for the larger carbon contents.

Li [29] noted that deformation occurred initially by dislocation motion but that at larger strains deformation twinning occurred. He noted that for the same strain, the deformation nano-twin density decreased with increasing carbon content due to the increase in stacking fault energy, the same observation as Stepanov et al. [21].

It is worth noting that when equiatomic CoCrFeMnNi with high carbon contents (0.5–2 at. %) are subject to cold-rolling followed by recrystallization, large volume fractions of carbides are often present as well as changes in grain size. The carbide's effect on the mechanical properties in addition to the reduction in solute strengthening from the reduction in carbon in solution [30–34] make it infeasible to determine the solid-solution strengthening effect of the carbon. CoCrFeMnNi with very high carbon contents (2.2–8.9 at. %) show large volume fractions of carbides even upon casting [28,35], but can still show very good $\varepsilon_f$ values.

Cheng et al. [36] added both titanium and carbon simultaneously to equiatomic CoCrFeMnNi making it impossible to determine the individual elemental contributions to the strength. Similarly, Klimova et al. [37] added both aluminum and carbon simultaneously to equiatomic CoCrFeMnNi, again making it impossible to determine the individual elemental contributions to the strength.

Klimova et al. [38] examined the effect of three different (measured) carbon levels (0.53, 0.95 and 2.11 at. %) on the strength of as-cast, large-grained (150–200 μm) non-equiatomic CoCrFeMnNi, i.e., CoCr$_{0.25}$FeMnNi, at both 77 K and 293 K: only the highest carbon level was reported to contain particles (<1 vol. %). Note that the undoped HEA tested for comparison contains a small amount of carbon (0.03 at. %). At least three specimens were tested for each composition. The carbon addition led to a linear increase in lattice parameter measured at room temperature from 0.3590 nm with no carbon to 0.3613 nm with 2.11 at. % C, a lattice parameter increase of 1.1 pm/at. % C or a lattice strain (= $\Delta a/(\Delta c \times a)$, were $a$ is the lattice parameter and $\Delta c$ is amount of interstitial) of 0.30 per at. %

C. A linear increase in $\sigma_y$ due to the carbon was found at both temperatures from 315 MPa with no carbon to 605 MPa with 2.11 at. % C at 77 K, and from 185 MPa with no carbon to 320 MPa with 2.11 at. % C at 293 K, see Figure 4a. These represent strengthening effects of 137 MPa/at. % C at 77 K and 64 MPa/at. % C at 293 K. It is worth noting that $\sigma_y$ for CoCr$_{0.25}$FeMnNi [38] is similar to the values reported at both 77 K and 293 K for equiatomic CoCrFeMnNi [27–29].

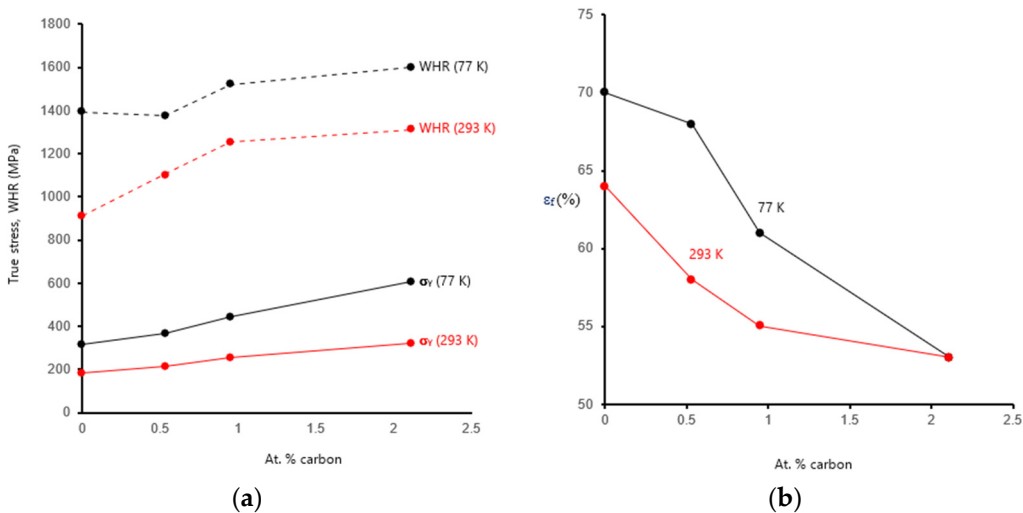

**Figure 4.** (**a**) Yield stress ($\sigma_y$) and work-hardening rate (WHR) for CoCr$_{0.25}$FeMnNi at 77 K and 293 K. (**b**) Elongation to failure ($\varepsilon_f$) for CoCr$_{0.25}$FeMnNi at 77 K and 293 K. Data from Klimova et al. [38].

The carbon addition produced a higher WHR (calculated from data in Table 2 [38]) at both temperatures, i.e., from 1376 to 1394 MPa for 0 and 0.53 at. % C to 1521 MPa for 0.95 at. % C at 77 K; and from 912 MPa for the undoped HEA, to 1105 MPa for 0.53 at. % C to 1255 MPa for 0.95 at. % C at 293 K, see Figure 4a. The values of WHR for CoCr$_{0.25}$FeMnNi are comparable to those for equiatomic CoCrFeMnNi noted above. Unlike the equiatomic CoCrFeMnNi, CoCr$_{0.25}$FeMnNi did not exhibit any signs of deformation twinning even at 77 K, but greater planarity of slip was observed at 77 K.

$\varepsilon_f$ decreased slightly with increasing carbon content at both temperatures, e.g., at 293 K from 64% with no carbon to 53% for 2.11 at. % carbon, see Figure 4b.

### 3.2. Carbon Doping—FeNiMnAlCr Alloys

Wang et al. [39] performed perhaps the most comprehensive study of the effects of an interstitial on large-grained (~150 μm), as-cast HEA specimens. They added 0, 0.07, 0.16, 0.30. 0.55 and 1.10 at. % C (measured) to Fe$_{40.4}$Ni$_{11.3}$Mn$_{34.8}$Al$_{7.5}$Cr$_6$ and determined the room temperature tensile properties, the lattice strain via synchrotron X-ray diffraction (XRD), the fracture behavior via scanning electron microscopy (SEM), and the dislocation structures via transmission electron microscopy. Atom probe tomography of the as-cast HEA containing 1.10 at. % C showed no nanoscale clustering or precipitation with a statistically homogeneous distribution of elements.

They found a linear relationship between the carbon content and both the change in lattice parameter and $\sigma_y$. The increase in lattice parameter was 2.74 pm/at. % C (or a lattice strain of 0.78 per at. % C), much larger than the 1.2 pm/at. % C and lattice strain of 0.30 per at. % C determined for CoCr$_{0.25}$FeMnNi [38]. This led to an increase in $\sigma_y$ of 184 MPa/at. % C, see Figure 5. The addition of carbon both lowered the SFE and increased the lattice friction stress. This produced a transition from wavy slip for the undoped HEA to planar slip for the carbon-doped HEAs at low strains. At high strains, the undoped HEA formed a cell structure, whereas a non-cell forming structure was found in carbon-doped HEAs. The formation of the non-cell forming structure, composed of a Taylor lattice, domain boundaries and microbands, produced an increase both in $\varepsilon_f$ from ~41% in the undoped HEA to ~51% in all the carbon-doped HEAs and in the WHR (calculated from data in Table 2 in [39]) from

1099 MPa in the undoped HEA to 2037 MPa in the 1.1 at. % C HEA, see Figure 5. The increase of WHR with increasing strain in carbon-doped HEAs delays the onset of necking, which further enhances $\varepsilon_f$.

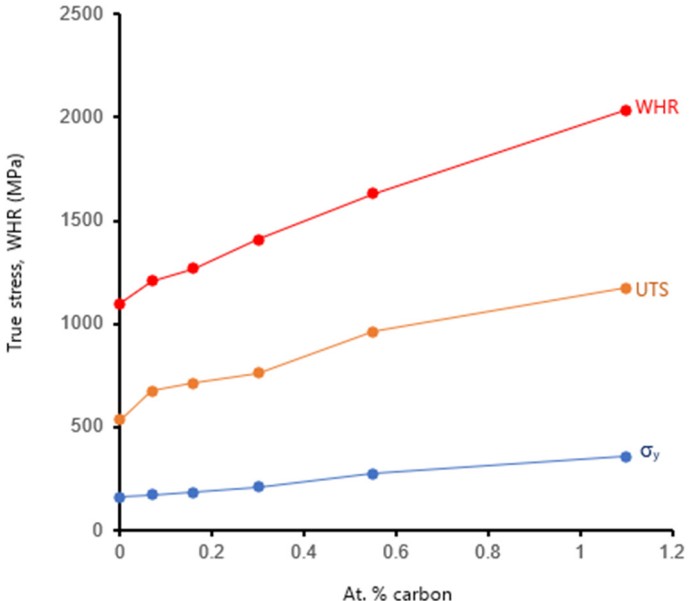

**Figure 5.** Yield stress ($\sigma_y$), ultimate tensile strength (UTS) and work-hardening rate (WHR) as a function of atomic percent carbon in $Fe_{40.4}Ni_{11.3}Mn_{34.8}Al_{7.5}Cr_6$. Data from Wang et al. [39].

Wang et al. [40] used a variety of thermo-mechanical treatments of the undoped and 1.1 at. % carbon-doped $Fe_{40.4}Ni_{11.3}Mn_{34.8}Al_{7.5}Cr_6$ to produce material that showed much higher yield strengths. However, the resulting material had a wide range of grain sizes and the undoped HEA contained ordered b.c.c. precipitates while the C-doped HEA contained carbides, making calculation of the interstitial strengthening problematic.

### 3.3. Carbon Doping—NiCoCr

While NiCoCr does not fit the definition of a HEA, it is worth considering because it is a f.c.c. MEA where several different levels of an interstitial have been added, i.e., Shang et al. [10] added varying amounts of carbon (0, 0.10, 0.25, 0.50 and 0.75 at. %) to equiatomic NiCoCr. The resulting alloys were single-phase, large-grained (140–160 μm) f.c.c. after a homogenization anneal, followed by cold-rolling and recrystallization anneal. The lattice parameter increased linearly with carbon concentration at a rate of 0.79 pm/at. % C. This is quite small compared to the 2.79 pm/at. % C found in $Fe_{40.4}Ni_{11.3}Mn_{34.8}Al_{7.5}Cr_6$ [39] or the 1.1 pm/at. % C found in $CoCr_{0.25}FeMnNi$ [38]. This gives a lattice strain of 0.22 per at. % C. Note that these authors also calculated the lattice strains on different planes from neutron diffraction measurements. The yield strength also increased linearly with carbon content from 242 MPa for the undoped MEA to 347 MPa for NiCoCr containing 0.75 at. % C, producing a strengthening rate of 140 MPa/at. % C. The authors fitted their data to a Labusch model [41], i.e., a $c^{2/3}$ dependence, although a $c^{1/2}$ dependence was almost as good a fit, suggesting that the strengthening arises from the elastic interaction of dislocations with the carbon atoms. Both the WHR and $\varepsilon_f$ were largely independent of carbon content, although there was a small increase in WHR and a very small decrease in ductility for the highest carbon content (0.75 at. %). Initially, deformation was solely by dislocation motion but twinning occurred at larger strains. This transition that was postponed by the addition of carbon, possibly due to the increase in stacking fault energy. They showed the stacking fault energy decreased with increasing carbon content, which delayed the onset of twinning and led to thinner twin bundles, but the accumulated dislocation density was increased.

### 3.4. Carbon Doping—Fe₄₀Mn₄₀Co₁₀Cr₁₀

*3.4. Carbon Doping—$Fe_{40}Mn_{40}Co_{10}Cr_{10}$*

Wei et al. [42] added 0.75 at. % and 1 at. % C to the as-cast large-grained (100–500 µm) single-phase medium entropy alloy (MEA) $Fe_{40}Mn_{40}Co_{10}Cr_{10}$. $\sigma_y$ for the undoped HEA was 240 MPa, which increased to 400 MPa and 520 MPa, respectively, with the two carbon additions, representing strength increases of 213 MPa/at. % C and 280 MPa/at. % C, see Table 3. The results imply an exponent for the concentration dependence of $\sigma_y$ greater than unity, which seems unlikely. It is not clear that more than one test was done for each alloy, and, thus, further testing might refine these values. XRD showed that the carbon increased the lattice parameter but the authors did not report the values. The WHR calculated from data in Table 1 of their paper shows that carbon increases the WHR by ~45% from 842 MPa for the undoped HEA to 1197 MPa and 1240 MPa for the 0.75 at. % C and 1 at. % C HEAs, respectively. $\varepsilon_f$ decreased slightly with increasing carbon addition from 56% when no carbon was present to 40% for an addition of 1 at. % C. These changes appeared to be related to greater deformation twinning relative to deformation by dislocation motion when carbon was added.

**Table 3.** The yield strength, $\sigma_y$, ultimate tensile strength, UTS, work-hardening rate, WHR, and elongation to failure, $\varepsilon_f$, for $Fe_{40}Mn_{40}Co_{10}Cr_{10}$ HEAs with various C contents. From Wei et al. [42] and Chen et al. [43].

| at. % Carbon | $\sigma_y$ (MPa) | UTS (MPa) | WHR (MPa) | $\varepsilon_f$ (%) | Reference |
|---|---|---|---|---|---|
| Undoped | 213 ± 7 | 471 ± 11 | 893 | 57.7 ± 2.5 | [43] |
| Undoped | 240 | 480 | 842 | 56 | [42] |
| 0.75 | 400 | 703 | 1197 | 44 | [42] |
| 1.0 | 520 | 818 | 1240 | 40 | [42] |
| 2.2 | 310 ± 6 | 650 ± 17 | 1135 | 62.9 ± 1.4 | [43] |
| 3.3 | 422 ± 7 | 787 ± 21 | 1128 | 77.8 ± 1.5 | [43] |

Chen et al. [43] explored the effects of much higher carbon concentrations (2.2, 3.3, 4.4, 6.6 and 8.9 at. %) on the room temperature mechanical properties of as-cast, large-grained (~95 µm) $Fe_{40}Mn_{40}Co_{10}Cr_{10}$. Three specimens were used for each alloy. XRD showed only f.c.c. peaks and the lattice parameter increased linearly up 4.4 at. % C, implying that carbon up to this level was in solution—oddly, larger carbon concentrations led to a decrease in the lattice parameter—however, carbides were noted in the HEA with 4.4 at. % C. Note that carbides were even noted in the HEA after hot forging at 1123 K. A dendritic structure was clearly evident for HEAs with ≥3.3 at. % C (Figure 2a in reference [43]), but was probably present for all compositions. The increase in lattice parameter per atomic percent carbon was 0.82 pm/at. %, yielding a lattice strain per at. % C, i.e., of 0.22/at. %, values which are very close to those observed by Shang et al. [10] for NiCoCr. Carbon additions to the as-cast HEA (which are in solution) up to 3.3 at. % C increased both $\sigma_y$ and the WHR (calculated from Table 1 in Chen et al. [43]), see Table 3. $\sigma_y$ increases roughly linearly with carbon concentration at ~63 MPa/at. % C, from 213 MPa for the undoped HEA to 422 MPa for the 3.3 at. % C HEA, which is much less than the value reported by Wei et al. [42]. Similarly, WHR increased from 893 MPa for the undoped HEA to ~1130 MPa for both the 2.2 at. % C and to 3.3 at. % C HEAs. Note that the increases in $\sigma_y$ and WHR obtained by Chen et al. [43] are either less than or comparable to the increases obtain by Wei et al. [42] for less than 25% of the carbon addition. The additions of carbon of 2.2 at.% and 3.3 at. % increased $\varepsilon_f$, see Table 3. Carbon's effects on the mechanical properties were attributed to the observation that the carbon suppressed dislocation motion, and, hence, promoted deformation twinning.

Li et al. [44] produced a carbon-doped mostly (>99%) f.c.c. HEA $Fe_{49.5}Mn_{30}Co_{10}Cr_{10}C_{0.5}$ that was cast, hot-rolled, given a homogenization anneal, cold rolled and recrystallized; the mechanical properties were compared with undoped $Fe_{50}Mn_{30}Co_{10}Cr_{10}$ produced in an earlier work [45]. However, while the carbon produced a large increase in strength, it is not possible to assess the interstitial strengthening

since the undoped alloy was 28% hexagonal-close-packed. Later, Zhang et al. [46] replaced an additional 1 at.% iron in this HEA with nitrogen to produce the f.c.c. HEA $Fe_{48.5}Mn_{30}Co_{10}Cr_{10}C_{0.5}N_{1.0}$ (at. %). The resulting HEA was even stronger than the HEA that was simply doped with carbon, but, again, it is not possible to assess the carbon or nitrogen strengthening effect since an undoped f.c.c. HEA is not available for comparison.

### 3.5. Nitrogen Doping

There have been no studies of the effects of nitrogen on HEAs where the nitrogen remained in solution. However, there have been studies on the effects of nitrogen on the MEAs NiCoCr and FeCoNiCr.

Moravcik et al. [11] hot-rolled ingots of both undoped NiCoCr and NiCoCr with 0.47 at. % nitrogen followed by a homogenization anneal for 2 h at 1200 °C to produce coarse-grained material. Some of this material was cold rolled to a 70% reduction and annealed at either 800 °C for 10 min or 30 min, or at 900 °C for 10 min to produce fine-grained material. Additional anneals were used to produce larger-grained MEAs. The chemistry, which was measured using inductively-coupled plasma mass spectroscopy, showed only small variations in both the major elements and in interstitials (oxygen, sulfur) between the undoped and N-doped MEA and that the oxygen and sulfur content were quite low. Atom probe tomography showed that only a random solid solution was present for N-doped MEA. Moravcik et al. [11] found that the 0.47 at. % N addition produced an increase in lattice parameter from 0.3568 to 0.3572 nm, which is a lattice strain of 0.24 per at. % N. This is quite close to the lattice strain of 0.22 per at. % C in NiCoCr, noted above [10].

The nitrogen increased the yield strength of the large-grained (43–45 µm) samples from 282 to 375 MPa (33%) with a small increase in UTS of 7% a minor increase in elongation to fracture from 76% to 79%, measured from three tensile tests on each MEA. Even the largest grain sizes tested are too small to exclude grain size strengthening effects, which are different in the doped and undoped MEA. The nitrogen had little effect on the WHR or $\varepsilon_f$ for these large grained specimens, which was 1614–1725 MPa. However, for nitrogen-doped MEA both the WHR and $\varepsilon_f$ decrease with decreasing grain size, see Figure 6.

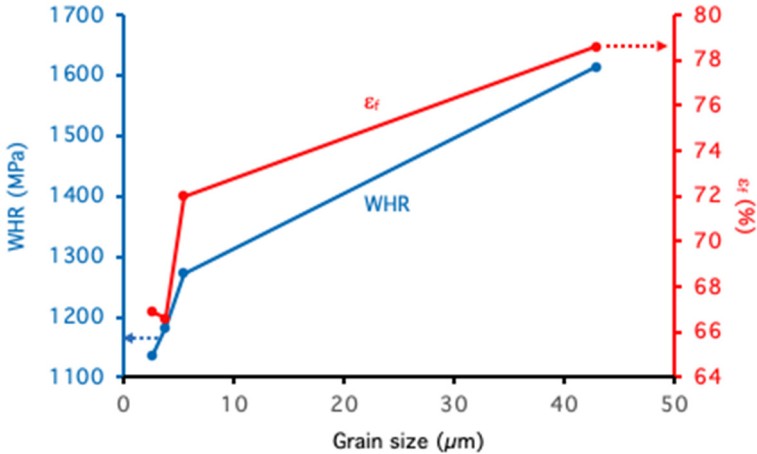

**Figure 6.** Work Hardening rate (WHR) and percent elongation to failure ($\varepsilon_f$) for NiCoCr MEA containing 0.47 at. % N. Data based on Table 2 in Moravcik et al. [11].

Moravcik et al. [11] determined the Hall–Petch relationship for the N-doped MEA and compared their $\sigma_0$ and $k$ values to those measured for the undoped MEA by Yoshida et al. [47]. Yoshida et al. [47] obtained a value of 216 MPa for $\sigma_0$ for the undoped MEA, which is similar to the value calculated from the critical resolved shear stress for a single crystal by Uzer et al. [48] of 211 MPa. The nitrogen addition increased $\sigma_0$ to 254 MPa [11]. This gives a strength increase of 81 MPa per at. % N, which is significantly less than the 140 MPa per at. % C in NiCoCr, noted above [10]. The effect of nitrogen on $k$ was more

dramatic than on $\sigma_0$ almost tripling it from 265 MPa $\mu$m$^{1/2}$ [47] to 748 MPa $\mu$m$^{1/2}$ [11]. Deformation occurred both by dislocation motion and deformation twinning but there was no quantification of the relative amounts of each.

Song et al. [12] added 1.83 at % nitrogen (measured) to the additively-manufactured MEA equiatomic FeCoNiCr. Atom probe tomography showed that the nitrogen (and other elements) were homogeneously distributed. The nitrogen was not found to produce nitrides, but led to a substantial increase in lattice parameter from 0.3524 to 0.3549 nm, indicative that a large amount of nitrogen was in solution. This increase corresponds to a lattice strain of 0.39 per at. % N. The nitrogen increased $\sigma_y$ from 520 to 650 MPa, while also producing an increase in $\varepsilon_f$ from 27% to 34%. However, the processing and nitrogen had a marked effect on the grain structure, producing a much finer structure. Thus, the increase in strength cannot be simply ascribed to the interstitial nitrogen.

The HEA FeNiMnAlCr was subjected to nitriding at elevated temperatures, which led to a substantial hardness increase at the surface, but the nitriding led to the removal of the aluminum from the matrix producing a change from a eutectic ordered b.c.c./f.c.c. lamellae microstructure to a f.c.c. matrix containing fine AlN particles [49]. A high concentration of nitrogen was also added to the MEA CrCoNi [50] but again resulted in the formation of second phases.

By doping with both carbon and nitrogen, Laurent-Brocq et al. [51] produced carbonitrides in CrMnFeCoNi, which substantially increased the hardness. While possibly technologically useful, it is not possible to determine the interstitial strengthening of carbon or nitrogen from their work.

### 3.6. Nitrogen/Oxygen Doping

Youssef at al. [52] mechanically alloyed batches of elemental powders to produce the lightweight HEA $Al_{20}Li_{20}Mg_{10}Sc_{20}Ti_{30}$. The result was a nanocrystalline (12 nm grain size), single-phase f.c.c. powders with a large lattice parameter of 0.4323 nm and a very high hardness of 5.8 GPa. One batch of the powders was contaminated with 0.4 at. % nitrogen and 1.39 at. % oxygen. The interstitials did not change the crystals structure or grain size but led to a slight increase in hardness to 6.1 GPa. Unfortunately, because both oxygen and nitrogen were present it is not possible to determine the contributions of each to the small strength increase.

### 3.7. Boron Doping

Hakan and Mohsen [53] produced a series of $Fe_{(52.7-x)}Mn_{(31.11)}Al_{5.09}Ni_{11.08}B_x$ ($x$ = 0, 0.05, 0.2, 0.5, 0.7 wt. %) medium entropy alloys that were hot-rolled, given a homogenization anneal, cold rolled and recrystallized. The boron was wholly in solution for additions up to 0.2 wt. %. These interstitial boron additions produced a small increase in strength (from 151 MPa for no boron to 170 MPa for 0.2 wt. % boron), but because they produced a substantial decrease in grain size (from 72 $\mu$m for no boron to 19 $\mu$m for 0.2 wt. % boron), it is not possible to separate the boron strengthening from the grain size strengthening.

Several other studies have added boron to various HEAs including to $Al_{0.5}CoCrCuFeNi$ [54], $Fe_{40.4}Ni_{11.3}Mn_{34.8}Al_{7.5}Cr_6$ [55], and AlFeCoNi [56]. All resulted in the formation of second phases.

### 3.8. Hydrogen Doping

Using an electrolytic technique, Luo et al. [57] introduced 8.01 wt. p.p.m. hydrogen into equiatomic CoCrFeMnNi that was cast, hot-rolled, given a homogenization anneal, cold rolled and recrystallized. Energy dispersive spectroscopy in a scanning electron microscope showed no large-scale segregation of elements. Interestingly, the hydrogen produced no change in $\sigma_y$, and very modest increases in both $\varepsilon_f$ and the ultimate tensile strength. The hydrogen was reported to increase deformation nano-twinning density.

## 4. Discussion

### 4.1. Lattice Strain

A major effect of interstitials is to cause an expansion of the crystal lattice. The HEAs and MEAs described above largely consist of transition metals and the f.c.c. unit cells are all around 0.36 nm. The HEAs and MEAs showed similar lattice strains of 0.22–0.39 for both carbon and nitrogen, see Table 4, except for $Fe_{40.4}Ni_{11.3}Mn_{34.8}Al_{7.5}Cr_6$. Interestingly, the lattice strain in the latter is significantly higher, which may be related to the fact that the HEA is the only one that contains a non-transition metal.

**Table 4.** Lattice strain, $\Delta a/(\Delta c \times a)$, and yield strength increase, $\Delta\sigma_y$, per atomic percent of various interstitials in different alloys from the references indicated.

| Alloy | Interstitial | $\Delta a/(\Delta c \times a)$ | $\Delta\sigma_y$ (MPa) | Reference |
|---|---|---|---|---|
| $Fe_{40}Mn_{40}Co_{10}Cr_{10}$ | C | 0.22 | 60 | [42] |
| NiCoCr | C | 0.22 | 140 | [10] |
| NiCoCr | N | 0.24 | 81 | [11] |
| $CoCr_{0.25}FeMnNi$ | C | 0.30 | 64 | [38] |
| FeCoNiCr | N | 0.39 | - | [12] |
| $Fe_{40.4}Ni_{11.3}Mn_{34.8}Al_{7.5}Cr_6$ | C | 0.78 | 184 | [39] |

### 4.2. Yield Strength

The effect of interstitials on the mechanical properties of a HEA can be most easily assessed for the effects of carbon on CoCrFeMnNi at room temperature because of the larger number of studies on this HEA than on other HEAs, see Table 2. Unfortunately, there is a huge range of values for the increase in $\sigma_y$ per at. % C on equiatomic CoCrFeMnNi at room temperature ranging from the zero strengthening reported by Li [29], to 55–75 MPa/at. % C (comparison of Stepanov et al. [21] and Sun et al. [19]; Chen et al. [28]), to 120 MPa/at.% C (comparison of Wu et al. [27] and Otto et al. [18]) to a high value of 163 MPa/at. % C (comparison of Stepanov et al. [21], and Otto et al. [18]). By comparison, a value of 64 MPa/at. % C was reported for non-equiatomic $CoCr_{0.25}FeMnNi$ [38]. Similarly, the values for the increase in $\sigma_y$ for CoCrFeMnNi at 77 K differ substantially from 145 MPa/at. % C [38] to 320 MPa/at. % (comparison of Wu et al. [27], and Otto et al. [18]).

A similar situation exists for $Fe_{40}Mn_{40}Co_{10}Cr_{10}$: whereas Wei et al. [42] obtained an increase of $\sigma_y$ of 213–280 MPa/at. % C, Chen et al. [28] found an increase of only 63 MPa/at. % C. Wei et al.'s [42] data indicate a concentration dependence of greater than one, whereas Chen et al. [28] obtained a value close to unity.

By comparison, Wang et al. [39] found an increase in room-temperature $\sigma_y$ of 184 MPa/at. % C for $Fe_{40.4}Ni_{11.3}Mn_{34.8}Al_{7.5}Cr_6$. The increase was directly related to the lattice expansion due to the C, and, again, $\sigma_y$ showed a linear concentration, $c$, dependence rather than the more commonly-observed $c^{1/2}$ or $c^{2/3}$ dependence.

The increase in $\sigma_y$ must be related to the lattice strain caused by the interstitial. However, the lattice strain alone is not sufficient to predict the increase in yield strength, as is evident by inspection of Table 4. For example, for NiCoCr, carbon and nitrogen both produce similar lattice strains of 0.22–0.24 but the increase in yield strength appears to be much greater for carbon (140 MPa versus 81 MPa).

It is interesting to compare the $\sigma_y$ increase due to carbon with that observed in non-HEA f.c.c. alloys. Carbon increases the strength of stainless steels by 77 MPa/at. % C [58], while carbon increases the strength of TWIP steels by only 42 MPa/at.% C [59,60]. Thus, carbon strengthens $Fe_{40.4}Ni_{11.3}Mn_{34.8}Al_{7.5}Cr_6$ more than steels. A comparison of steels with CoCrFeMnNi and $Fe_{40}Mn_{40}Co_{10}Cr_{10}$ is much less clear because of the wide range of values obtained for these HEAs.

### 4.3. Work-Hardening Rate

While the increase in $\sigma_y$ must be related to the lattice strain caused by the interstitial atoms, the change in work-hardening rate is also affected by how the interstitial affects the lattice friction and the stacking fault energy, which affect the planarity of slip and cross-slip, and the onset of twinning.

For undoped equiatomic CoCrFeMnNi the WHR is independent of grain size or increases slightly with increasing grain size for grain sizes >1 μm [18–20,28], see Figure 7. Below a critical grain size of ~1–2 μm there may be a sharp reduction in WHR. For 1 at. % C doped equiatomic CoCrFeMnNi there appears to be an increase in WHR with increasing grain size [21], see Figure 7. We do not include the value for the finest grain size material produced by Stepanov et al. [21] because of the very inhomogeneous microstructure and the wide range of $\sigma_y$ values, although we note that the stress-strain curve for this HEA shows little work-hardening (Figure 9 in reference [21]).

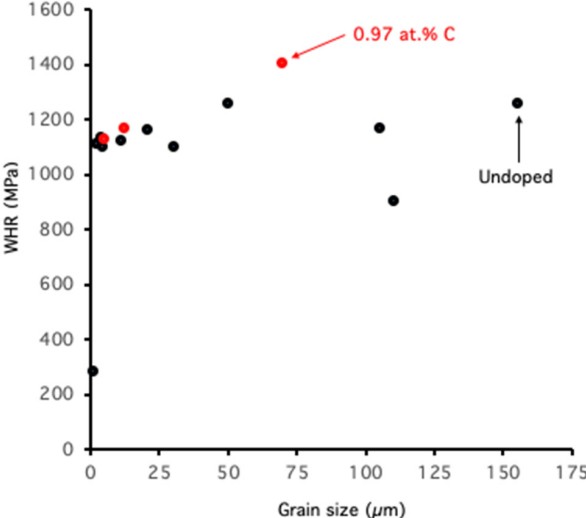

**Figure 7.** Work Hardening Rate (WHR) at room temperature versus grain size for undoped equiatomic CoCrFeMnNi—black points—(calculated from [18–20,28]), and for 0.97 at. % carbon doped CoCrFeMnNi—red points (calculated from [21]).

Regarding the effect of carbon on WHR, while Wu et al. [27] and Klimova et al. [38] found that 0.5–2.0 at. % C increased the WHR of CoCrFeMnNi at both 77 K and 293 K by about 20%, Li [29] found only a small increase in WHR for 0.25–0.9 at. % C and Chen et al. [28] found 1.1 at. % C had little effect at 293 K.

We should note that Stepanov et al. [21] and Li [29] reported that carbon reduced the extent of deformation twinning while Wu et al. [27] and Chen et al. [28] reported that carbon increased the extent of deformation twinning in equiatomic CoCrFeMnNi. Note that Klimova et al. [38] found that deformation twinning did not occur in $CoCr_{0.25}FeMnNi$ with or without carbon.

For large-grained NiCoCr, neither nitrogen nor carbon had much effect on the WHR [10,11]. However, as noted earlier, the WHR decreased with decreasing grain size for the nitrogen-doped MEA [11] see Figure 6, as was noted for both carbon-doped [21] and undoped CoCrFeMnNi [18–20,28]. As for equiatomic CoCrFeMnNi, initial deformation in NiCoCr was solely by dislocation motion but twinning occurred at larger strains, a transition that was delayed by the addition of carbon [10]. This effect was ascribed to the observed decrease in stacking fault energy with increasing carbon content, which not only delayed the onset of twinning and led to thinner twin bundles, but also increased the accumulated dislocation density.

Both Wei et al. [42] and Chen et al. [43] found substantial increases in WHR for carbon additions from 0.75-3.3 at. % for $Fe_{40}Mn_{40}Co_{10}Cr_{10}$, a feature probably related to the greater deformation twinning observed with carbon.

Wang et al. [39] found both an increase (up to 62%) in WHR and that the WHR increased to higher strains in $Fe_{40.4}Ni_{11.3}Mn_{34.8}Al_{7.5}Cr_6$ for up to 1.1 at. % C. These features were related to the change in

deformation behavior from initially wavy slip at low strains for the undoped HEA to cell formation at high strains, whereas for the C-doped HEAs planar slip was observed at low strains and a non-cell forming structure composed of the Taylor lattice, domain boundaries and microbands was found at high strains in the C-doped HEAs.

### 4.4. Ductility

For undoped equiatomic CoCrFeMnNi the room temperature ductility appears to increase slightly with increasing grain size for grain size >3–4 μm; below 3–4 μm $\varepsilon_f$ decreases rapidly [18–20,28], see Figure 8. A similar critical grain size dependence is present for 1 at. % C doped equiatomic CoCrFeMnNi, but the transition grain size appears to be larger at around 12 μm [21], see Figure 8. This is reminiscent of the behavior observed for the medium-entropy eutectoid alloy $Fe_{28}Ni_{18}Mn_{33}Al_{21}$, where $\varepsilon_f$ increases with increasing lamellar spacing until around 1 μm after which it is independent of lamellar spacing [61].

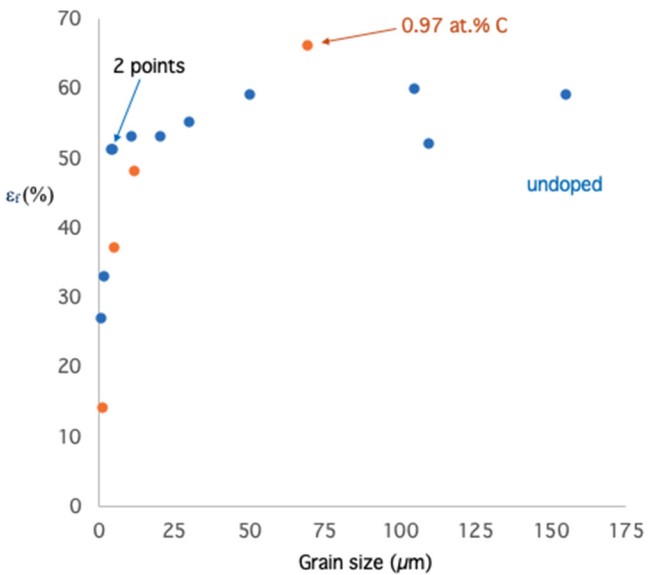

**Figure 8.** Elongation to fracture ($\varepsilon_f$) at room temperature versus grain size for undoped equiatomic CoCrFeMnNi—black points (data from [18–20,28]), and for 0.97 at. % carbon doped CoCrFeMnNi—red points (data from [21]).

Chen et al. [28] indicated that an improvement in $\varepsilon_f$ occurred on addition of carbon to CoCrFeMnNi, but some other studies obtained similar values of $\varepsilon_f$ to Chen et al. [28] with C-doped CoCrFeMnNi in undoped material. Other studies [27] suggest a large decrease in $\varepsilon_f$ both at 77 K and 293 K due to carbon, whereas some work suggests that carbon has little effect [29].

Although Wei et al. [42] and Chen et al. [43] obtained very similar $\varepsilon_f$ values (56–58%) for undoped $Fe_{40}Mn_{40}Co_{10}Cr_{10}$, data on the effects of carbon are again mixed. Wei et al. [42] found a decrease in $\varepsilon_f$ when 1 at. % C was added to $Fe_{40}Mn_{40}Co_{10}Cr_{10}$, whereas Chen et al. [43] found an increase in $\varepsilon_f$ for carbon additions up to 3.3 at. %.

For large-grained NiCoCr, neither nitrogen nor carbon had much effect on $\varepsilon_f$ [10,11]. However, $\varepsilon_f$ decreased with decreasing grain size for the nitrogen-doped MEA [11], see Figure 6.

The situation is clearer for large-grained $Fe_{40.4}Ni_{11.3}Mn_{34.8}Al_{7.5}Cr_6$ where Wang et al. [39] found a 20% increase in $\varepsilon_f$ due to the addition of C. This increase occurred for the addition of only 0.07 at. % C and further additions up to 1.1. at. % C made little difference.

*4.5. Hall–Petch Slope*

The effect of carbon on the Hall–Petch slope can be ascertained for equiatomic CoCrFeMnNi by comparing the results between different researchers, i.e., 1 at. % C increased the room temperature value of $k$ from 394 to 586 MPa·μm$^{-1/2}$ [18–20] to 935 MPa·μm$^{-1/2}$ [21]. The effect of 0.47 at. % nitrogen on $k$ for the equiatomic MEA NiCoCr was more striking almost tripling it from 265 MPa·μm$^{1/2}$ [47] to 748 MPa·μm$^{1/2}$ [11]. While this is a remarkable increase, by comparison, only 60 p.p.m. C added to high-purity b.c.c. iron increased $k$ by more than a factor of five (from 100 MPa·μm$^{1/2}$ to 550 MPa·μm$^{1/2}$, whereas nitrogen additions made little difference [62]. Additionally, ageing of an interstitial-free steel containing 50 p.p.m. carbon for 60 ks at 373 K produced an increase in $k$ from 522 MPa·μm$^{1/2}$ to 771 MPa·μm$^{1/2}$ as the carbon segregated to the grain boundaries [63].

The increase in $k$ for equiatomic CoCrFeMnNi and for equiatomic NiCoCr suggests that the interstitials segregate to the grain boundary and makes slip transfer across the grain boundary more difficult although no measurements of this were undertaken.

## 5. Conclusions

This paper has analyzed the literature on the effects of interstitials on the mechanical properties of single phase f.c.c. HEAs. While there is substantial scatter in the data in the literature, there are a number of points that can be made:

1.  Carbon increases the yield strength at both room temperature and 77 K. This increase is greater than in traditional alloys for $Fe_{40.4}Ni_{11.3}Mn_{34.8}Al_{7.5}Cr_6$. However, the data for CoCrFeMnNi and $Fe_{40}Mn_{40}Co_{10}Cr_{10}$ are scattered and it is not clear whether the strengthening is greater than for traditional alloys.
2.  The yield strength increase arises from the lattice strain produced by the interstitial, but there is no simple correlation between the lattice strain and the yield strength increase, indicating that other factors are also important.
3.  There is a weak grain size dependence of both the ductility and work-hardening rate in both undoped and carbon-doped CoCrFeMnNi for grain sizes greater than the order of a few microns with smaller grain sizes showing the lower values. Below ~1 μm there is rapid decrease in ductility for both undoped and carbon-doped CoCrFeMnNi, and, at least for the undoped HEA, the WHR shows a sharp decrease at grain sizes <2 μm. Nitrogen-doped NiCoCr also showed a decrease in both ductility and WHR with decreasing grain size [11]. This is a phenomenon worth examining in other HEAs, and may be related to a change in deformation mechanism at small grain sizes as seen in $Fe_{40.4}Ni_{11.3}Mn_{34.8}Al_{7.5}Cr_6$ [40].
4.  The effects of carbon on ductility are less clear. In many studies, carbon produced small decreases in elongation or little change while in others an increase in ductility has been observed. Thus, the effects may be system specific and depend on how carbon influences the deformation mechanisms.
5.  The effects of carbon on the work-hardening are also unclear. Some studies indicate the insubstantial effect of carbon on the work-hardening rate whereas some have shown large increases, which are related to changes in deformation behavior.
6.  Carbon almost doubles the Hall–Petch strengthening in CoCrFeMnNi (the only material for which data is available), suggesting substantial carbon segregation to the grain boundaries. Thus, carbon can be an effective strengthener both by increasing the lattice resistance and the grain boundary strength.

There have been few studies on the effects of other interstitials such as boron, nitrogen and hydrogen, which have been shown to be potent strengtheners in other traditional alloys. It is evident both that interstitials can be potent strengtheners and that more research is needed on interstitials to understand their effects on mechanical properties and to optimize their use. To do so, researchers need to measure the interstitial content after casting, rather than assuming that the interstitial content is

the same as that added to the melt. The grain size and, possibly, texture have to be considered when comparing the data from a HEA with and without an interstitial.

**Acknowledgments:** This research was supported by the US Department of Energy, Office of Basic Energy Sciences Grant No. DE-SC0018962. The author would like to thank Bram Kuijer, Rachel Osmundsen and Andrew Peterson for their insightful comments on the manuscript.

**Conflicts of Interest:** The author declares no conflict of interest.

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
