# Peer review of "Interstitials in f.c.c. High Entropy Alloys"

_metals, doi:10.3390/met10050695_

Round 1
Reviewer 1 Report
The author systematically studied the effects of interstitial atoms on the mechanical properties of fcc HEAs. Research studies from different groups on the strengthening effects due to the interstitial atoms are included. Several parameters related to the mechanical properties are also summarized in this review article, e.g. the lattice resistance, elongation, the Hall-patch slope, the work-hardening rate. However, some additional comments should be further addressed.
The author should be careful with the data when carbon no longer exist as interstitial atoms but in the format of carbide. These data cannot be used for the extrapolation of the lattice resistance. For example, the author concluded that the yield strength increases with carbon additions at a rate of ~60 MPa/at.% C from 213 MPa (undoped HEA) to 467 MPa (4.4 at.% C doped HEA) in line 375 page 11. However, after a careful check of the source article (Ref. 32), the dendrite structure formed for the C content exceeding 3.3 at.%. Thermal-mechanical treatment towards the 3.3 at.% C specimen induced the formation of carbides, which suggest that carbon no longer existed as interstitial atoms. As a result, the reliability to conclude the effect of interstitial atoms according to such data should be reduced.
Besides, the author should also summarise the solubility of carbon among HEAs from literatures to find out the threshold between interstitial atoms and carbide formation and then compare it with traditional alloys.
It appears that the increment of yield strength due to carbon doping is generally higher than that of conventional alloys, though not all the data support this hypothesis. Is there any explanation for it? Or does it depend on the specific base alloy systems?
The author claimed that the higher Hall-Petch slope after carbon-doping can be ascribed to the strong carbon segregation to the grain boundaries which make the dislocation slip between different grains more difficult. Is there any direct experimental observation for the carbon segregation at the grain boundaries?
Some other studies on interstitial atoms are not mentioned in this article. For example, Intermetallics 106 77-87 (2019).
Moreover, there are several typos in this manuscript:
In line 16 page 1, two repeated "room temperature" in the abstract.
In line 85 page 3, there is an extra "has".
In line 375 page 11, wrong reference number "32".
Author Response
The author systematically studied the effects of interstitial atoms on the mechanical properties of fcc HEAs. Research studies from different groups on the strengthening effects due to the interstitial atoms are included. Several parameters related to the mechanical properties are also summarized in this review article, e.g. the lattice resistance, elongation, the Hall-patch slope, the work-hardening rate. However, some additional comments should be further addressed.
The author should be careful with the data when carbon no longer exist as interstitial atoms but in the format of carbide. These data cannot be used for the extrapolation of the lattice resistance. For example, the author concluded that the yield strength increases with carbon additions at a rate of ~60 MPa/at.% C from 213 MPa (undoped HEA) to 467 MPa (4.4 at.% C doped HEA) in line 375 page 11. However, after a careful check of the source article (Ref. 32), the dendrite structure formed for the C content exceeding 3.3 at.%. Thermal-mechanical treatment towards the 3.3 at.% C specimen induced the formation of carbides, which suggest that carbon no longer existed as interstitial atoms. As a result, the reliability to conclude the effect of interstitial atoms according to such data should be reduced.
Response: We have revised the discussion in several places in the text to include only carbon additions up to 3.3 at.%. We have also noted the dendritic structures and the carbides present in the HEA with 4.4 at. % C. Interestingly, the exclusion of the data for the 4.4 at.% C makes only a minor difference to the conclusions.
Besides, the author should also summarise the solubility of carbon among HEAs from literatures to find out the threshold between interstitial atoms and carbide formation and then compare it with traditional alloys.
Response: This is a good point, but there is insufficient data in the literature. In some cases, we know when particles are present, but that does not provide the solubility limit. Similarly, when someone has a HEA with X in solution, that simply sets the lower limit to the solubility. Further, Many HEAs are not in the equilibrium state.
It appears that the increment of yield strength due to carbon doping is generally higher than that of conventional alloys, though not all the data support this hypothesis. Is there any explanation for it? Or does it depend on the specific base alloy systems?
The author claimed that the higher Hall-Petch slope after carbon-doping can be ascribed to the strong carbon segregation to the grain boundaries which make the dislocation slip between different grains more difficult. Is there any direct experimental observation for the carbon segregation at the grain boundaries?
Response: This is a good point. We have added the following to clarify this point: “The effect of 0.47 at. % nitrogen on k for the equiatomic MEA NiCoCr was more striking almost tripling it from 265 MPa.µm1/2 [U] to 748 MPa.µm1/2 [Y]. While this is remarkable increase, only 60 p.p.m. C added to high-purity b.c.c. iron increased k by more than a factor of five (from 100 MPa.µm1/2 to 550 MPa.µm1/2, whereas nitrogen additions made little difference [Q]. Also, ageing of an interstitial-free steel containing 50 p.p.m. carbon for 60 ks at 373 K produced an increase in k from 522 MPa.µm1/2 to 771 MPa.µm1/2 as the carbon segregated to the grain boundaries [P].
The increase in k for equiatomic CoCrFeMnNi and for equiatomic NiCoCr suggest that the interstitials segregate to the grain boundary and makes slip transfer across the grain boundary more difficult although no measurements of this were undertaken.
Some other studies on interstitial atoms are not mentioned in this article. For example, Intermetallics 106 77-87 (2019).
Response: We have added a substantial section on the suggested paper: Y.Y. Shang, Y. Wu, J. Y. He, X. Y. Zhu, S. F. Liu, H. L. Huang, K. An, Y. Chen, S. H. Jiang, H. Wang, X. J. Liu and Z. P. Lu, “Solving the strength-ductility tradeoff in the medium-entropy NiCoCr alloy via interstitial strengthening of carbon”, Intermetallics. 106 (2019) 77-87, and also included comments in the Absract, Discussion and Conclusions.
Moreover, there are several typos in this manuscript:
In line 16 page 1, two repeated "room temperature" in the abstract. Removed
In line 85 page 3, there is an extra "has". Removed
In line 375 page 11, wrong reference number "32". Changed to “43”
Reviewer 2 Report
This paper deals with f.c.c. High Entropy Alloys and reports in details the interstitials (carbon, boron, nitrogen or hydrogen doping) effect on mechanical properties of HEA’s materials. Author mainly focused on carbon doping effect. The rest interstitials were described shortly. It must be considered the presented paper is written in logical way, good in English and can be interesting for both, science and industry communities. However the research documentations and results discussion should be presented as review form. Due to objective and clear evaluation of this manuscript is hard to made.
Author Response
I don’t understand the comment that “the research documentations and results discussion should be presented as review form. Due to objective and clear evaluation of this manuscript is hard to made.”. I don’t know what is meant by “review form”. I also ran this comment by a co-worker and they could not understand it either.
Reviewer 3 Report
The manuscript reviews the existing literature about the influence of interstitial elements added to fcc-high entropy alloys on their mechanical properties, yield stress (YS) and work hardening rate (WHR), from 77 K up to 1073 K. This survey reveals that such topic has been barely studied. Moreover, clear conclusions cannot be deduced from many of these works because of inhomogeneous microstructures, segregations precipitation of second phases...... This is perfectly revised in this manuscript. However, it is missing a detailed analysis of the mechanisms governing the deformation of these alloys. These mechanisms are different depending on the temperature and they determine YS and WHR of the alloys. This should be clearly stated in the review, making clear that changes in absolute values can be due to changes in the deformation mechanism. At low temperatures 77 and 273 K twinning/nanotwinning predominates while at high temperatures dislocation motion controls deformation. In addition, it would be interesting to analyze the data existing in the literature to correlate simultaneous effect of carbon (solute) content and/or grain size on the deformation mechanisms. Moreover, it will be interesting to understand if changes in the ratio of the elements constituting the alloy could influence YS and WHR. This will complete the manuscript.
Author Response
The manuscript reviews the existing literature about the influence of interstitial elements added to fcc-high entropy alloys on their mechanical properties, yield stress (YS) and work hardening rate (WHR), from 77 K up to 1073 K. This survey reveals that such topic has been barely studied. Moreover, clear conclusions cannot be deduced from many of these works because of inhomogeneous microstructures, segregations precipitation of second phases...... This is perfectly revised in this manuscript. However, it is missing a detailed analysis of the mechanisms governing the deformation of these alloys. These mechanisms are different depending on the temperature and they determine YS and WHR of the alloys. This should be clearly stated in the review, making clear that changes in absolute values can be due to changes in the deformation mechanism. At low temperatures 77 and 273 K twinning/nanotwinning predominates while at high temperatures dislocation motion controls deformation.
Response: The paper already describes the deformation mechanisms in several cases. We have added to this and described the deformation behavior for all alloys when it has been reported in the papers reviewed.
In addition, it would be interesting to analyze the data existing in the literature to correlate simultaneous effect of carbon (solute) content and/or grain size on the deformation mechanisms. This will complete the manuscript.
Response: In several cases we have already made an attempt to separate the effects of interstitial content from grain size and have discussed this, e.g. Figures 6-8 (Figure 7 is a new figure), but this is not possible in all HEAs because of the lack of data.
Moreover, it will be interesting to understand if changes in the ratio of the elements constituting the alloy could influence YS and WHR.
Response: Only in one case is the effect of changing the constituent elements on the HEA for the same interstitial been done, i.e. for equiatomic CoCrFeMnNi and non-equiatomic CoCrFeMnNi, and we added some additional comments to describe those effects.
I think the reviewer’s comments really point to the fact that more work needs to be done on the effects of interstitials.
Reviewer 4 Report
Review contains literature overview on the influence of carbon addition on mechanical properties of HEAs. The topic is important for the community and will be interesting for readers. Author discusses mainly mechanical properties but microstructure and phase composition were out of the discussion. Nevertheless, I am sure that mechanical properties should be correlated with microstructure, morphology and phase composition. It is important to discuss in which form (solid solution, carbide nanoparticles etc) carbon, nirtogen, boron are presented. It will give much more clear overview of the problem. I suggests to show more microscopic and structural information and discuss it.
There are a couple of recent papers discussing a structural aspects of microadditions in multicomponent alloys and HEAs:
Riva et al Int.Materials Rev. 2019 V. 61(3) 203
Riva et al J. Alloys Comp. 2018 V. 730 544
There are everal minor comments:
I suggest to combine figeres 2 and 3.
Table 3: why some lines were marked in red?
Author Response
Review contains literature overview on the influence of carbon addition on mechanical properties of HEAs. The topic is important for the community and will be interesting for readers. Author discusses mainly mechanical properties but microstructure and phase composition were out of the discussion. Nevertheless, I am sure that mechanical properties should be correlated with microstructure, morphology and phase composition. It is important to discuss in which form (solid solution, carbide nanoparticles etc) carbon, nirtogen, boron are presented. It will give much more clear overview of the problem. I suggests to show more microscopic and structural information and discuss it.
Response: WE documented this information already for each alloy. To make this clearer, we added the sentence that “We note that unless otherwise stated below, the interstitials are assumed to be completely in solution.”.
There are a couple of recent papers discussing a structural aspects of microadditions in multicomponent alloys and HEAs:
Riva et al Int.Materials Rev. 2019 V. 61(3) 203
Response: This is review paper on Scandium effects in alloys and is, thus, not relevant.
Riva et al J. Alloys Comp. 2018 V. 730 544
Response: This paper is not on a f.c.c. HEA but on a B2 HEA and does not consider the effect of solute but of adding particulate reinforcing elements, and is, thus, not relevant.
There are several minor comments:
I suggest to combine figeres 2 and 3.
Response: I considered this when submitting the paper, but I think it will be too messy.
Table 3: why some lines were marked in red?
Response: The use of red and black was used to attribute the papers to authors. We have removed the red and added another column that shows the attribution for the data.
Reviewer 5 Report
The review manuscript is well written and contains a survey of available data that can be of interest to many readers. There are only small issues which should be addressed, as described bellow
1 -This statement is not correct: "There have been no studies in which a HEA with and without an interstitial have been undertaken" There is a recent study which directly compared the CoCrNi alloy with the same grain sizes with and without interstitials:https://doi.org/10.1016/j.scriptamat.2019.12.007
The said article can be also used to extend section on Nitrogen doping in the manuscript, which is rather limited. Nitrogen is shown to be better for strengthening than carbon in, for instance, austenitic steels due to better microstructural stability (nitride formation is slower than the formation of carbides).
It may be of interest to include the data from this work in the review
2 -The author should consider including two more issues in the review paper on this topic which significantly influence the values of K coefficient and friction stress in all studies:
a) Different authors use different threshold misorientation (from EBSD) values for defining grain boundaries
Some authors use minimum value of 5 degrees, whereas others even use 15 degrees. This changes the parameters in H-P equation considerably
b) There is a problem with annealing twin boundaries. In some alloys with low SFE such as CoCrNi or Cantor alloy, annealing twin boundaries can comprise over 50% of total grain boundary fraction. Some authors include them in the grain size measurement since they do increase strength, other authors tent to exclude them since it was done so traditionally. This produces significant discord in the data and interpretation. issue There is a great paper regarding this https://doi.org/10.1016/j.ijplas.2019.08.009
3 - The author compares the available data on different alloys in terms of work hardening rate (WHR). This value is however extremely dependent on grain size, so comparing different materials in this way is not completely valid. The author should consider using some other approach - like the normalization of WHR by grain size or other measures.
4 - The author uses a study (Effect of thermomechanical processing on microstructure and mechanical properties of the carbon-containing CoCrFeNiMn high entropy alloy) to show that K coefficient is over 900 MPa/um. This value is too large, which can be partially caused by high dislocation density and the presence of carbides.
Author Response
1 -This statement is not correct: "There have been no studies in which a HEA with and without an interstitial have been undertaken" There is a recent study which directly compared the CoCrNi alloy with the same grain sizes with and without interstitials:https://doi.org/10.1016/j.scriptamat.2019.12.007
The said article can be also used to extend section on Nitrogen doping in the manuscript, which is rather limited. Nitrogen is shown to be better for strengthening than carbon in, for instance, austenitic steels due to better microstructural stability (nitride formation is slower than the formation of carbides).
It may be of interest to include the data from this work in the review
Response: The statement is correct since CoCrNi is NOT a high entropy alloy. However, we have included discussion of the paper suggested by the reviewer and added the phrase “- although studies have been conducted on the medium entropy alloys NiCoCr with and without carbon [X] or nitrogen [Y] and on FeCoNiCr with and without nitrogen [42]” to the quoted sentence.
2 -The author should consider including two more issues in the review paper on this topic which significantly influence the values of K coefficient and friction stress in all studies:
a) Different authors use different threshold misorientation (from EBSD) values for defining grain boundaries
Some authors use minimum value of 5 degrees, whereas others even use 15 degrees. This changes the parameters in H-P equation considerably
Response: This is a very good point and we have added the sentence “Different authors use different threshold misorientations to determine the grain size from electron backscatter pattern data obtained in a scanning electron microscope, e.g. some use 5o misorientation while some use 15o, which can result in quite different grain sizes.”
b) There is a problem with annealing twin boundaries. In some alloys with low SFE such as CoCrNi or Cantor alloy, annealing twin boundaries can comprise over 50% of total grain boundary fraction. Some authors include them in the grain size measurement since they do increase strength, other authors tent to exclude them since it was done so traditionally. This produces significant discord in the data and interpretation. issue There is a great paper regarding this https://doi.org/10.1016/j.ijplas.2019.08.009
Response: This is an excellent point, although it is impossible to determine grain sizes in papers discussed based on removing twin boundaries. We have added the sentence “Schneider et al. [Z] recently showed that for the MEA NiCoCr that if twin boundaries are included in the Hall-Petch analysis that the k values can be lower (~600 MPa.µm-1/2 versus ~820 MPa.µm-1/2 at 273 K), although the temperature dependence of k is similar.”
3 - The author compares the available data on different alloys in terms of work hardening rate (WHR). This value is however extremely dependent on grain size, so comparing different materials in this way is not completely valid. The author should consider using some other approach - like the normalization of WHR by grain size or other measures.
Response: The comment is well taken. We already make the point in the paper that WHR depends on grain size. Indeed Figure 6 is included to make this point. We have also added a new Figure 7 that further illustrates this point. It is not possible to normalize the WHR for grain size since the functionality is not known.
4 - The author uses a study (Effect of thermomechanical processing on microstructure and mechanical properties of the carbon-containing CoCrFeNiMn high entropy alloy) to show that K coefficient is over 900 MPa/um. This value is too large, which can be partially caused by high dislocation density and the presence of carbides.
Response: This is not correct. Figure 7 in the quoted paper shows a dislocation-free matrix and the carbides present are negligible (≤0.2%). To make this point more clearly we modified the text to read “containing (measured) 0.97 at. % C that was given a homogenization anneal, 80% cold-rolled and annealed at temperatures from 873-1373 K. Specimens annealed at temperatures ≥973 K were fully recrystallized and occasionally contained a very small volume fraction (≤0.2%) of M23C6 particles within grains that would have had a negligible effect on the mechanical properties.”
Round 2
Reviewer 2 Report
Dear Author,
You previously selected "Article" as a type of your paper. However it looks like Review paper. I am sorry for misunderstanding. Now I have no more comments. Best regards,
Reviewer
Reviewer 4 Report
author addressed comments from reviewers.